# A generalized theoretical framework to investigate multicomponent actin dynamics

**Mintu Nandi**[1¤], **Shashank Shekhar** [2]*, **Sandeep Choubey** [3,4]*

**1** Department of Chemistry, Indian Institute of Engineering Science and Technology, Shibpur, Howrah, India, **2** Departments of Physics, Cell Biology and Biochemistry, Emory University, Atlanta, Georgia, United States of America, **3** The Institute of Mathematical Sciences, CIT Campus, Taramani, Chennai, India, **4** Homi Bhabha National Institute, Training School Complex, Anushaktinagar, Mumbai, India

¤ Current address: Universal Biology Institute, The University of Tokyo, Bunkyo-ku, Tokyo, Japan
* shekhar@emory.edu (SS); sandeep@imsc.res.in (SC)

**Data availability statement:** All the analysis scripts used in this study are available on GitHub at https://github.com/Mintu-Nandi-24/generalized-theory-of-actin.

## Abstract

The length of actin filaments is regulated by the combined action of hundreds of actin-binding proteins. While the roles of individual proteins are well understood, how they combine to regulate actin dynamics *in vivo* remains unclear. Recent advances in microscopy have enabled precise, high-throughput measurements of filament lengths over time. However, the absence of a unified theoretical framework has hindered a mechanistic understanding of the multicomponent regulation of actin dynamics. To address this, we propose a general kinetic model that incorporates the combined effects of an arbitrary number of regulatory proteins on actin dynamics. We derive exact closed-form expressions for the moments of (1) the distribution of filament lengths over time and (2) the long-time distribution of changes in filament lengths within a fixed time window. We show that these moments allow us to distinguish between different regulatory mechanisms of multicomponent regulation of actin dynamics. Our theoretical framework provides a powerful tool for interpreting existing data and guiding future experiments.

## Author summary

Actin filaments are essential components of cells, playing key roles in processes like cell movement, division, wound healing, and shape maintenance. The length of actin filaments inside cells is tightly controlled by the collective action of many actin-binding proteins. While the effects of individual proteins have been studied in great detail, how multiple proteins work together to regulate filament dynamics in living cells remains poorly understood. Recent improvements in microscopy allow precise, high-throughput measurements of filament lengths over time, offering new avenues to explore this question. In this work, we present a general mathematical framework that captures the

**Funding:** MN acknowledges SERB, India, for the National Post-Doctoral Fellowship [PDF/2022/001807]. SS was funded by NIH NIGMS Grant No. R35GM143050. SC acknowledges the support provided by the DBT Ramalingaswami Fellowship. The funders had no role in study design, data collection and analysis, decision to publish, or preparation of the manuscript.

combined effects of multiple regulatory proteins on actin filament dynamics. We derive exact expressions for various statistical properties of filament lengths such as the different moments, including how these properties evolve over time and how filament lengths change within a fixed time window in the long time limit. We show that these mathematical results can help identify and distinguish between different mechanisms of multicomponent regulation. This framework provides a valuable tool for interpreting experimental data and designing future studies on actin dynamics in complex cellular environments.

## 1. Introduction

Actin filaments are essential components of the cytoskeleton, driving various cellular processes such as motility, cytokinesis, endocytosis, and wound healing, etc [1–3]. The proper regulation of actin filament length is crucial for the execution of these functions and is achieved through the coordinated action of a variety of actin-binding proteins (ABPs) that modulate filament elongation, depolymerization, and capping [4–10]. Decades of biochemical and biophysical studies have shed light on the individual effects of many of these regulators. However, how multiple ABPs collectively shape actin dynamics in cells remains an open question.

Recent advances in high-resolution fluorescence microscopy have enabled precise measurements of changes in filament length over time for hundreds of filaments, producing rich, time-resolved datasets on filament length distributions [11–14], as illustrated in Fig 1. Similar insights into filament dynamics *in vivo* have been made possible by quantitative live-cell imaging [15]. Theoretical models have played a vital role in interpreting these experiments, providing mechanistic insights into actin filament assembly and turnover [14–19]. Broadly, two main classes of modeling efforts have emerged: one class focusing on the long-time regulation of filament length [17–20], and another addressing the time-dependent aspects of filament growth and turnover [21,22].

For instance, Mohapatra et al. [20] proposed the "antenna mechanism" to explain length control of actin cables in yeast, where filament length is regulated through the interplay of formin, Smy1, and myosin motors. Banerjee et al. demonstrated that competition between formin and capping proteins can generate bimodal length distributions at steady state [18], while Johann et al. [19] showed how molecular motors can regulate filament length by balancing growth and shrinkage rates. Time-resolved models have also explored the influence of ATP–actin and ADP–Pi caps on elongation dynamics [22], as well as the role of regulators such as twinfilin in shaping early and intermediate filament length distributions [21]. Despite these advances, most existing theoretical models have been limited to scenarios involving one or a small number of ABPs. As a result, the field currently lacks a general theoretical framework—a "theory of the experiment" [26]—that can connect filament length distributions to underlying molecular mechanisms in systems with multicomponent regulation. This gap limits our ability to uncover general principles governing actin filament dynamics in physiologically relevant cellular environments.

To address this gap, we introduce a general and analytically tractable theoretical framework for modeling actin filament dynamics under the simultaneous regulation of multiple ABPs. Unlike earlier approaches that either focused on long-time behavior or required computationally intensive simulations [18,20], our framework captures both (1) the time evolution of filament length distributions (Fig 1A) and (2) the long-time behavior of changes in filament length within a fixed time window (Fig 1B). Here, long-time behavior refers to the

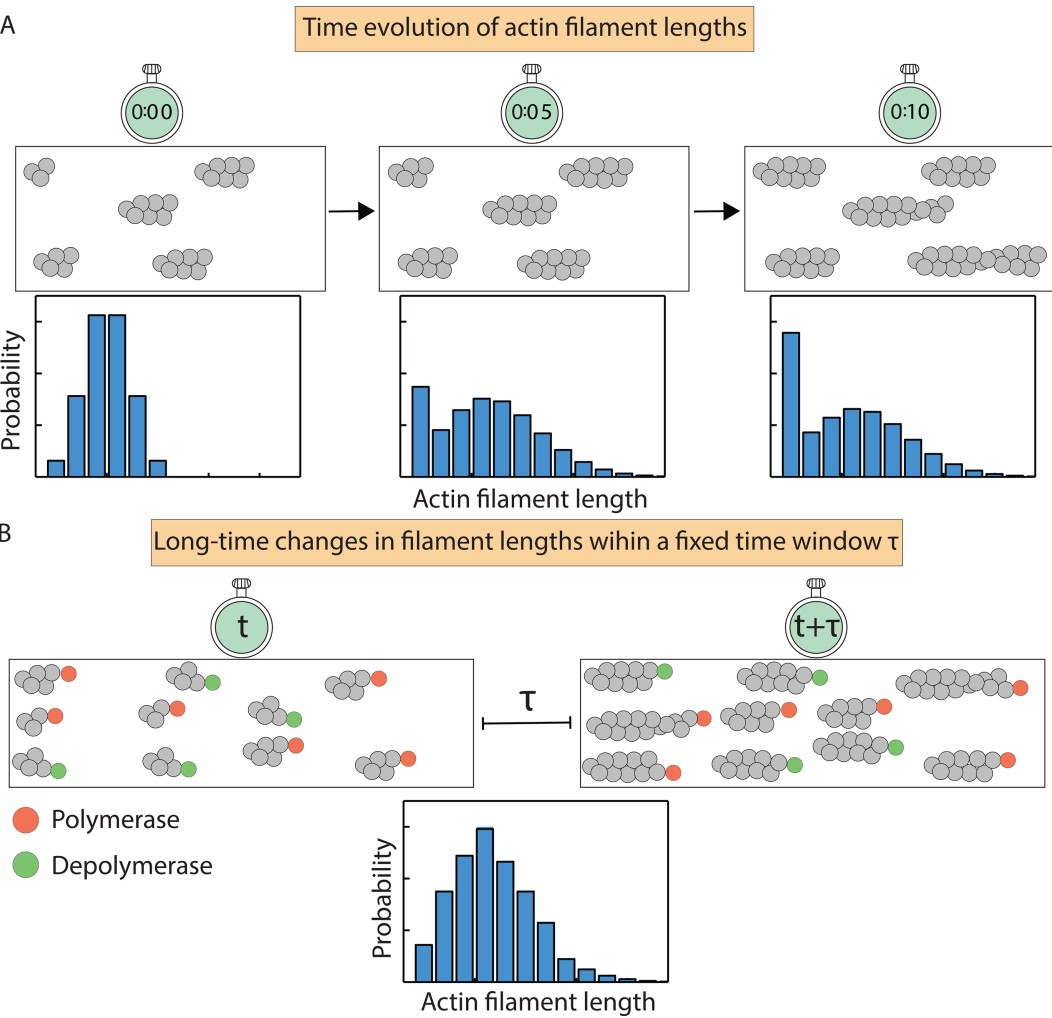

**Fig 1. Time evolution of actin filament lengths.** (A) A schematic illustrating experimental time traces of actin filament growth, providing snapshots of the length distribution at consecutive time intervals [23–25]. (B) Experimental time traces of filament length in the long-time limit, providing the distribution of length changes measured over a fixed time window $\tau$. Within this window, the fraction of filaments bound to each regulatory protein remains constant on average in the log-time regime, reflecting a dynamic stationary state.

regime where state probabilities become time-invariant, while the fixed time window represents experimentally relevant intervals over which changes in filament length are measured. The analytical tractability of our model allows for the exact computation of different statistical properties such as the mean, variance, and Fano factor, thereby enabling systematic discrimination between different regulatory scenarios and providing interpretable predictions for both transient and long-term dynamics. As a case study, we apply this framework to examine the combined effects of two ABPs with contrasting mechanisms: an elongator [27] and a capper [28]. Elongators such as formins bind filament barbed ends and accelerate polymerization in the presence of profilin–actin complexes, while cappers such as capping protein bind barbed ends and halt elongation. Although these regulators were previously believed to act in a mutually exclusive fashion, recent work has shown that both can simultaneously occupy the same filament end, leading to complex elongation dynamics [29,30]. Our framework

incorporates these interactions, offering exact analytical predictions for filament length statistics and serving as a predictive tool for designing experiments to distinguish between alternative length control mechanisms.

We present the mathematical formulation of our model, derive exact expressions for mean filament length change and the Fano factor, and apply the framework to explore how different regulatory regimes shape filament growth variability. Our results highlight the predictive power of analytical approaches for dissecting complex regulatory interactions in actin dynamics.

## 2. Methods

We develop a general kinetic model for multicomponent regulation of actin filament dynamics, where each filament stochastically transitions between discrete states. These states represent different combinations of bound actin-binding proteins (ABPs), determining whether the filament polymerizes, depolymerizes, or remains capped. We note that our model does not explicitly account for polymerization or depolymerization from the barbed or pointed ends, and is agnostic to the precise binding locations of ABPs. The filament can occupy a total of $N$ distinct states, divided into two classes: $N_1$ polymerizing states and $N_2 = N - N_1$ depolymerizing states. Capped states are included within the polymerizing states by assigning them a polymerization rate of zero. Alternatively, capped states could equivalently be grouped with the depolymerizing states without any loss of generality. In our model, each of the $N_1$ polymerizing states is associated with a polymerization rate $r_i$ ($i = 1, \dots, N_1$), while each of the remaining $N_2$ depolymerizing states is characterized by a depolymerization rate $\gamma_j$ ($j = N_1 + 1, \dots, N$). The filament switches between states at rates $k_{ij}$, representing transitions from state $j$ to state $i$. The model involves two random variables– the state of the filament $i$, and the change in its length, $\Delta L_t$, over a given time interval ($t$). The change in length is defined as $\Delta L_t = L_t - L_0$, where $L_t$ is the filament's length at time $t$ and $L_0$ is its initial length at $t = 0$. $\Delta L_t$ can be positive or negative depending on whether polymerization or depolymerization cumulatively dominates at time $t$. A positive $\Delta L_t$ indicates overall growth, while a negative value indicates net shrinkage. We note that this does not imply the filament is polymerizing or depolymerizing at time $t$; rather, it reflects the net balance of growth and shrinkage events at that time. We have presented a simple kinetic model in S1 Text to show the negative mean growth for pure depolymerization. This unified kinetic scheme allows us to model filament dynamics under diverse regulatory scenarios.

We focus on $P_i(\Delta L_t)$, which describes the probability that filament length has changed by $\Delta L_t$ while the filament is in state $i$ at time $t$. The time evolution of $P_i(\Delta L_t)$ across all states is governed by the following master equation in matrix form,

$$\frac{d}{dt}\mathbf{P}(\Delta L_t) = (\hat{\mathbf{K}} - \hat{\mathbf{R}})\mathbf{P}(\Delta L_t) + \hat{\mathbf{R}}\left[\mathbf{P}_\uparrow(\Delta L_t - 1) + \mathbf{P}_\downarrow(\Delta L_t + 1)\right], \tag{1}$$

where $\mathbf{P}(\Delta L_t) = (P_1(\Delta L_t), \cdots, P_{N_1}(\Delta L_t), P_{N_1+1}(\Delta L_t), \cdots, P_N(\Delta L_t))^T$ denotes the vector of probabilities across states.

To obtain analytical solutions of Eq (1), we introduce separate probability vectors for polymerization and depolymerization (Fig A in S1 Text). Specifically, $\mathbf{P}_\uparrow(\Delta L_t - 1)$ represents the probabilities of polymerizing states, and $\mathbf{P}_\downarrow(\Delta L_t + 1)$ represents the probabilities of depolymerizing states. The elements in $\mathbf{P}_\uparrow$ are zero for depolymerizing states, and vice versa for $\mathbf{P}_\downarrow$ (Fig A in S1 Text). The matrix $\hat{\mathbf{K}}$ in the master Eq (1) describes state transitions, with off-diagonal elements $k_{ij}$ representing the rate of transition from state $j$ to $i$, and diagonal

elements $k_{ii}$ representing the outflow rate from state $i$ (Fig A in S1 Text). The matrix $\hat{\mathbf{R}}$ is diagonal, representing the rates of polymerization and depolymerization (Fig A in S1 Text). From Eq (1), we derive exact expressions for the moments of 1) the distribution of $\Delta L_t$ over time, and 2) the long-time distribution of $\Delta L_\tau$ within a fixed time window $\tau$, where long-time limit implies that the probability distribution of $\Delta L_\tau$ ceases to change. In this regime, the transitions between different states of actin filament become stationary, allowing the statistics of $\Delta L_\tau$ to reflect time-invariant behavior. In Fig 2A and 2B, we schematically illustrate a representative example of this general framework, showing a three-state model of regulation of actin-filament length and its corresponding master equation.

We now calculate the $n$th moment of the distribution of $\Delta L_t$ by multiplying both sides of Eq (1) by $\Delta L_t^n$. By summing over all values of $\Delta L_t$ and finally multiplying both sides by $\vec{Y} = (1, 1, \cdots, 1)_{1 \times N}$, as described in S1 Text, we obtain the following equation for the $n$-th moment,

$$\langle \Delta L_t^n \rangle \;=\; \vec{R}\left[\vec{\Lambda}_{t_\uparrow}^{(0)} + (-1)^n \vec{\Lambda}_{t_\downarrow}^{(0)}\right] + \vec{R}\sum_{x=1}^{n-1}\binom{n}{x}\left[\vec{\Lambda}_{t_\uparrow}^{(x)} + (-1)^{n+x}\vec{\Lambda}_{t_\downarrow}^{(x)}\right], \tag{2}$$

A

<div align="center">

**A simple model of multicomponent regulation of actin length**

</div>

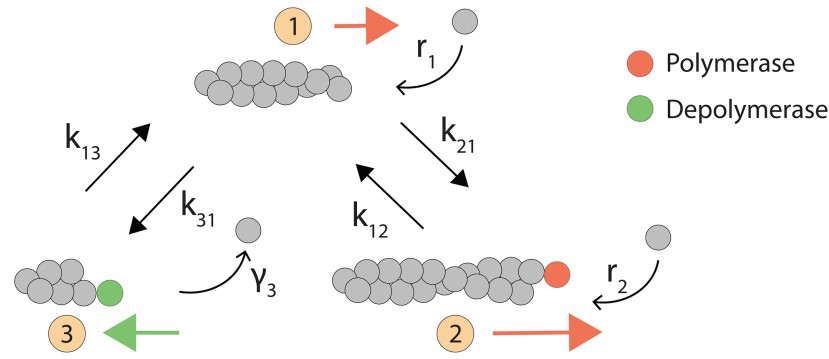

B

<div align="center">

**Governing equation of actin length**

</div>

$$\frac{d}{dt}\mathbf{P}(\Delta L_t) \;=\; (\hat{\mathbf{K}} - \hat{\mathbf{R}})\mathbf{P}(\Delta L_t) + \hat{\mathbf{R}}\left[\mathbf{P}_\uparrow(\Delta L_t - 1) + \mathbf{P}_\downarrow(\Delta L_t + 1)\right]$$

$$\mathbf{P}(\Delta L_t) = \begin{pmatrix} P_1(\Delta L_t) \\ P_2(\Delta L_t) \\ P_3(\Delta L_t) \end{pmatrix} \quad \hat{\mathbf{K}} = \begin{pmatrix} -(k_{21} + k_{31}) & k_{12} & k_{13} \\ k_{21} & -k_{12} & 0 \\ k_{31} & 0 & -k_{13} \end{pmatrix} \quad \hat{\mathbf{R}} = \begin{pmatrix} r_1 & 0 & 0 \\ 0 & r_2 & 0 \\ 0 & 0 & \gamma_3 \end{pmatrix}$$

$$\mathbf{P}_\uparrow(\Delta L_t - 1) = \begin{pmatrix} P_1(\Delta L_t - 1) \\ P_2(\Delta L_t - 1) \\ 0 \end{pmatrix} \quad \mathbf{P}_\downarrow(\Delta L_t + 1) = \begin{pmatrix} 0 \\ 0 \\ P_3(\Delta L_t + 1) \end{pmatrix}$$

**Fig 2. Model of multicomponent regulation of actin filament length.** (A) Effect of different regulatory proteins on actin dynamics, illustrated through a three-state kinetic model. (B) Master equation describing the evolution of the filament length distribution, along with the different matrices characterizing the master equation.

where, $\vec{\Lambda}_{t_\uparrow}^{(0)} = \mathcal{L}^{-1}\left[\overrightarrow{\Delta L}_{s_\uparrow}^{(0)}/s\right]$, $\vec{\Lambda}_{t_\downarrow}^{(0)} = \mathcal{L}^{-1}\left[\overrightarrow{\Delta L}_{s_\downarrow}^{(0)}/s\right]$, $\vec{\Lambda}_{t_\uparrow}^{(x)} = \mathcal{L}^{-1}\left[\overrightarrow{\Delta L}_{s_\uparrow}^{(x)}/s\right]$, and $\vec{\Lambda}_{t_\downarrow}^{(x)}$

$= \mathcal{L}^{-1}\left[\overrightarrow{\Delta L}_{s_\downarrow}^{(x)}/s\right]$ (see S1 Text for details). Here, $\mathcal{L}^{-1}$ denotes the inverse Laplace transforma-

tion of the partial moment vectors $\overrightarrow{\Delta L}_{s\cdots}^{(\cdots)}$ defined in Laplace space $s$. We note here that $\vec{\Lambda}$s are obtained as functions of the matrices $\hat{\mathbf{K}}$, and $\hat{\mathbf{R}}$.

Next, we calculate the $n$-th moment of the distribution of $\Delta L_\tau$ in the long-time limit within the time window $\tau$. This log-time regime is defined by the condition that the time derivative of the state occupancy distribution is zero, which corresponds to the stationary regime (see S1 Text). Practically, this condition can be realized by allowing the system to evolve for a sufficiently long duration ($t \to \infty$), ensuring that the state occupancies converge to their stationary values. Observable changes in filament length are then recorded over the finite observation window $\tau$. The $n$th moment over this window is given by

$$\langle \Delta L_\tau^n \rangle = \vec{R}\left[\overrightarrow{\Delta L}_\uparrow^{(0)} + (-1)^n \overrightarrow{\Delta L}_\downarrow^{(0)}\right]\tau + \vec{R}\sum_{x=1}^{n-1}\binom{n}{x}\left[\vec{\Lambda}_{\tau_\uparrow}^{(x)} + (-1)^{n+x}\vec{\Lambda}_{\tau_\downarrow}^{(x)}\right], \qquad (3)$$

where, $\vec{\Lambda}_{\tau_\uparrow}^{(x)} = \mathcal{L}^{-1}\left[\dfrac{\overrightarrow{\Delta L}_{s_\uparrow}^{(x)}\big|_\tau}{s}\right]$, $\vec{\Lambda}_{\tau_\downarrow}^{(x)} = \mathcal{L}^{-1}\left[\dfrac{\overrightarrow{\Delta L}_{s_\downarrow}^{(x)}\big|_\tau}{s}\right]$ (see S1 Text for details). Here, $\overrightarrow{\Delta L}_{s\cdots}^{(x)}\big|_\tau$

denotes the long-time partial moment vectors. In the above equation, $\overrightarrow{\Delta L}_{\cdots}^{(0)}$ stands for the zeroth order partial moment vectors, which characterize the occupancy of various states. The $\vec{\Lambda}$s are obtained as functions of the the matrices $\hat{\mathbf{K}}$, and $\hat{\mathbf{R}}$.

We note that the $n$th moment of filament growth in both transient and long-time limit (Eqs (2) and (3), respectively) are expressed in the unit of subunits$^n$ of actin monomers. However, for $n = 1$, both $\langle \Delta L_t \rangle$ and $\langle \Delta L_\tau \rangle$ are referred to as mean filament growth throughout the manuscript. This quantity represents the average change in filament length relative to an initial reference length, where the averaging is performed over a population of actin filaments.

The analytical close-form expressions of $n$th moment distinguish this framework from prior approaches. To demonstrate the utility of our theoretical framework, we dissect specific regulatory mechanisms of actin dynamics. We employ our analytical results to distinguish between different mechanisms of multicomponent regulation of actin dynamics involving multiple ABPs.

## Results

### Regulation of actin filament length by a single ABP

To explore how filament growth distributions reveal mechanistic insights into multicomponent regulation of actin dynamics, we examine the combined effects of an elongator and a capper protein on filament length. Elongators promote growth [4,5], while cappers inhibit polymerization [6]. We consider three scenarios: 1) actin filaments with only an elongator, 2) actin filaments with only a capper, and 3) actin filaments with both proteins. In the ensuing sections, we discuss these scenarios in detail.

**Effect of an elongator on actin filament length.** We first examine how an elongator affects actin filament length. In the presence of an elongator, the filament can exist in two states: a bare state (B) and an elongator-bound state (BF), with corresponding polymerization rates $r_1$ and $r_2$, respectively, (see (Fig 3A)). The binding and unbinding rates of the elongator are $k_F^+ = \widetilde{k}_F^+[F]$ and $k_F^-$, where $[F]$ is the elongator concentration. The mean and variance of $\Delta L_t$, as a function of time and the different biochemical rates, are given by

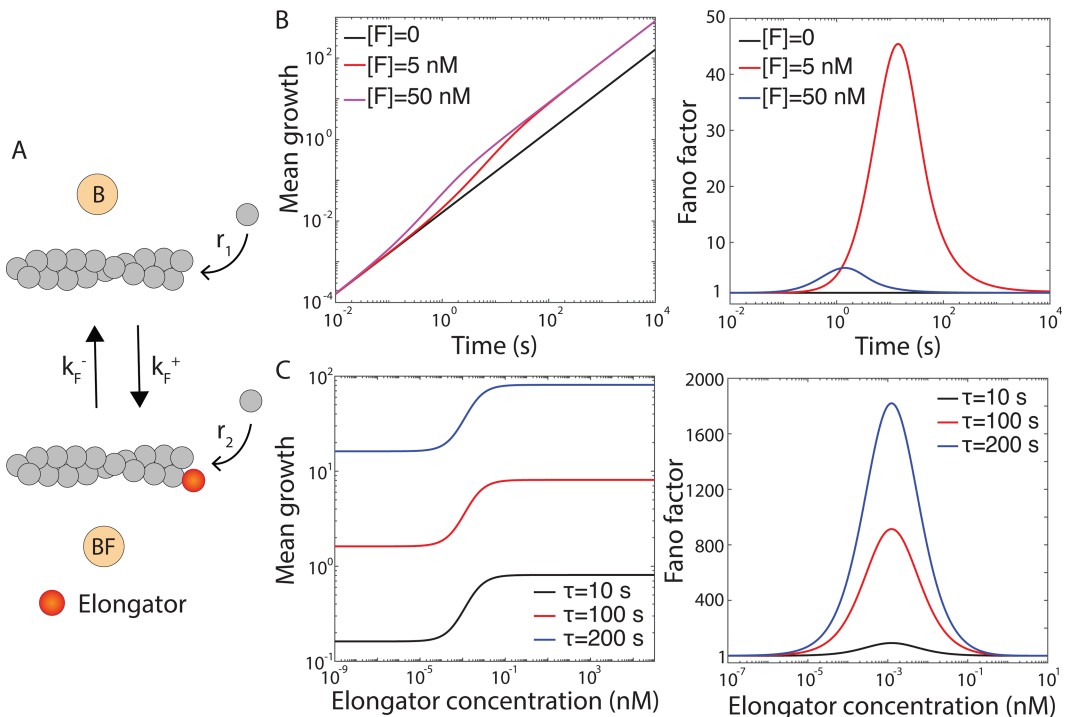

**Fig 3. Effect of an elongator on actin filament length.** (A) Two-state model of actin polymerization in the presence of an elongator is illustrated. (B) Mean filament growth ($\mu m$) and Fano factor are plotted as a function of time for varying elongator concentrations. (C) long-time mean growth ($\mu m$) and Fano factor are shown as functions of elongator concentration for different $\tau$ values. To convert the mean growths from subunits to $\mu m$, we use 1 subunit $\sim 0.0027 \mu m$ [23]. However, we keep Fano factor in the unit of subunits of actin monomers. Following parameters are used for elongator (formin): $r_1 = 6$ subunits/s, $r_2 = 30$ subunits/s, $\widehat{k_F^+} = 29.1 \, \mu M^{-1} s^{-1}$, and $k_F^- = 8.1 \times 10^{-5} \, s^{-1}$ [23].

$$\langle \Delta L_t \rangle \quad = \quad r_2 t + (r_1 - r_2) A_F t + \frac{(r_1 - r_2)(1 - A_F)}{D_F}(1 - e^{-D_F t}), \tag{4}$$

$$\begin{aligned}
\sigma_{\Delta L_t}^2 \quad = \quad & r_2 t + (r_1 - r_2) A_F t + \frac{2 A_F (1 - A_F)(r_1 - r_2)^2}{D_F} t \\
& + \frac{(r_1 - r_2)(1 - A_F)\left[D_F + (r_1 - r_2)(1 - 5 A_F)\right]}{D_F^2} - \left[\frac{(1 - A_F)(r_1 - r_2)}{D_F}\right]^2 e^{-2 D_F t} \\
& + \frac{(1 - A_F)(r_1 - r_2)}{D_F^2}\left[4 A_F (r_1 - r_2) - D_F\{1 + 2(1 - 2 A_F)(r_1 - r_2) t\}\right] e^{-D_F t}, \tag{5}
\end{aligned}$$

where $D_F = k_F^+ + k_F^-$ and $A_F = k_F^- / D_F$. When $[F] = 0$, actin filament length increases linearly over time, driven solely by the polymerization rate of the bare (B) state (Fig 3B). In the presence of the elongator, filament growth initially remains linear but becomes nonlinear at intermediate times due to the transition to the BF state. In the long time limit, growth is governed by the polymerization rate corresponding to the BF state only. The Fano factor (defined as the ratio of variance and mean), which quantifies filament-to-filament variability in growth, remains at one in the absence of an elongator ($[F] = 0$). In the presence of an elongator, it rises above one, peaks, and then decreases, approaching one (Fig 3B). The intermediate rise in the Fano factor indicates a mixed population of filaments in the B and BF states, which depends strongly on elongator concentration.

Next, we explore the properties of the long-time distribution of $\Delta L_\tau$ for a population of actin filaments. The closed-form expressions for the long-time mean and variance of $\Delta L_\tau$ within a fixed time window $\tau$, are given by

$$
\begin{aligned}
\langle \Delta L_\tau \rangle &= r_2\tau + (r_1 - r_2)A_F\tau, & (6)\\
\sigma^2_{\Delta L_\tau} &= r_2\tau + (r_1 - r_2)A_F\tau + \frac{2}{D_F}(r_1 - r_2)^2 A_F(1 - A_F)\tau \\
&\quad - \frac{2}{D_F^2}(r_1 - r_2)^2 A_F(1 - A_F)(1 - e^{-D_F\tau}). & (7)
\end{aligned}
$$

In the long-time limit, mean filament growth increases with elongator concentration, plateauing at high concentrations (Fig 3C). The Fano factor exhibits non-monotonic behavior: it remains at one at low concentrations, rises above one, peaks, and then decreases, approaching one at high concentrations (Fig 3C). The increase at intermediate concentrations is due to filaments switching between the B and BF states. The convergence of Fano factor to 1 at both extremes of elongator concentration reflects the fact that the system effectively behaves as a one-state process (Poisson) in these limits. For example, when the elongator concentration is very low, the filament predominantly resides in the bare (B) state; conversely, at very high elongator concentrations, the filament dynamics is dominated by the elongator-bound (BF) state. This behavior can also be understood from the limiting behavior of Eqs (6) and (7). When the elongator concentration is very low (i.e., $[F] \to 0$), $D_F \approx k_F^-$ and $A_F \approx 1$. In this limit, the mean filament growth becomes $\langle \Delta L_\tau \rangle \approx r_1\tau$, and the variance also becomes $\sigma^2_{\Delta L_\tau} \approx r_1\tau$, leading to Fano factor $\approx 1$. Similarly, at very high elongator concentration, $D_F \gg 1$ and $A_F \ll 1$ and the mean growth becomes $\langle \Delta L_\tau \rangle \approx r_2\tau$ and variance becomes $\sigma^2_{\Delta L_\tau} \approx r_2\tau$, again giving Fano factor $\approx 1$.

**Effect of a capper on actin filament length.** Next, we examine how a capper protein affects actin filament length. In this model (see Fig 4A), the filament can exist in two states: a bare state (B) and a capper-bound state (BC), with polymerization rates $r_1$ and $r_2 = 0$, respectively. The capper binding and unbinding rates are $k_C^+ = \widetilde{k}_C^+[C]$ and $k_C^-$. The mean and variance of the distribution of $\Delta L_t$ are given by

$$
\begin{aligned}
\langle \Delta L_t \rangle &= r_1 A_C t + \frac{r_1(1 - A_C)}{D_C}(1 - e^{-D_C t}), & (8)\\
\sigma^2_{\Delta L_t} &= r_1 A_C t + \frac{r_1(1 - A_C)}{D_C}(1 - e^{-D_C t}) + \frac{2r_1^2 A_C(1 - A_C)t}{D_C} \\
&\quad + \left[\frac{r_1(1 - A_C)}{D_C}\right]^2 (1 - e^{-2D_C t}) - \frac{2r_1^2(1 - A_C)}{D_C^2}(1 - e^{-D_C t}) \\
&\quad + \frac{2r_1^2(1 - A_C)(1 - 2A_C)}{D_C^2}\left[1 - (1 + D_C t)e^{-D_C t}\right], & (9)
\end{aligned}
$$

where $D_C = k_C^+ + k_C^-$ and $A_C = k_C^-/D_C$. At $[C] = 0$, the mean growth is higher than when $[C]>0$ (Fig 4B). As capper concentration increases, the growth rate decreases, demonstrating an inverse relationship between $[C]$ and filament growth (Fig 4B).

When $[C] = 0$, the Fano factor is one, and filament growth is Poissonian (Fig 4B). Adding capper initially does not affect growth variability, as it does not bind to the barbed end. Over time, the capper binds, capping the filament and increasing the Fano factor, with a more pronounced increase at lower concentrations. The transient Fano factor exhibits such behavior

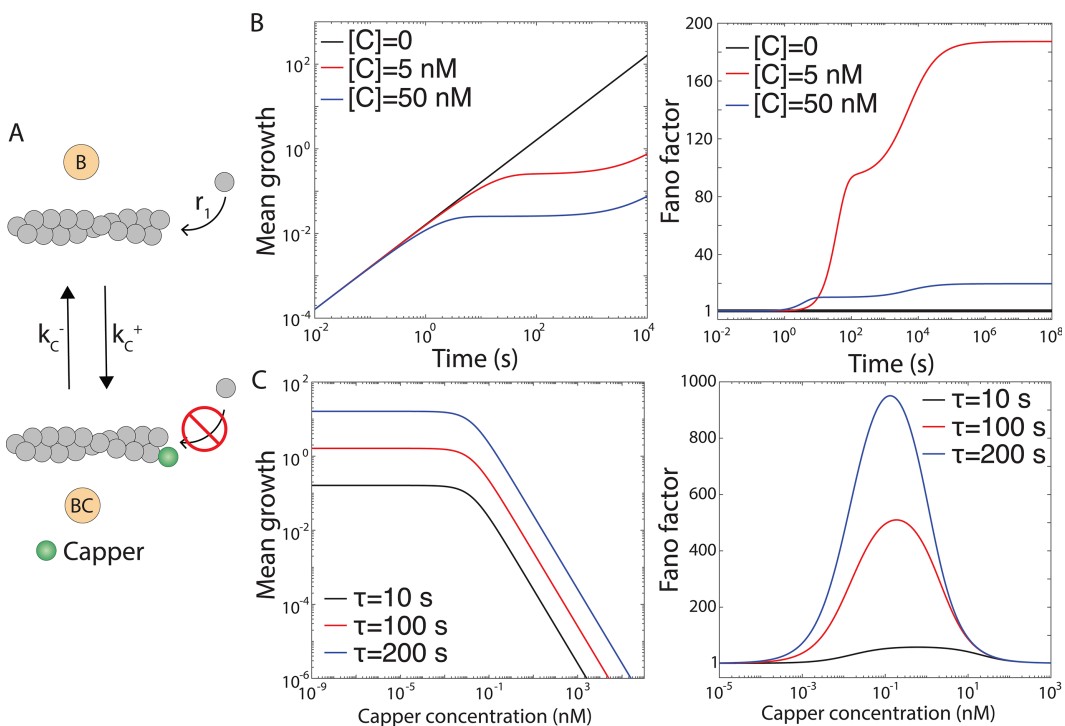

**Fig 4. Effect of a capper on actin filament length.** (A) Two-state model of actin dynamics in the presence of a capper is shown. (B) Mean filament growth ($\mu m$) and Fano factor are plotted as functions of time for different capper concentrations. (C) long-time mean growth ($\mu m$) and Fano factor are shown as functions of capper concentration for different values of $\tau$. To convert the mean growths from subunits to $\mu m$, we use 1 subunit $\sim 0.0027\mu m$ [23]. However, we keep Fano factor in the unit of subunits of actin monomers. Following parameters are used for the capper(Capping protein): $r_1 = 6$ subunits/s, $\widetilde{k}_C^+ = 12.8\,\mu M^{-1}s^{-1}$, and $k_C^- = 2.0 \times 10^{-4}\ s^{-1}$ [23].

due to frequent transitions between the bare (B) and capped (BC) states. In the long-time limit, the Fano factor reaches a plateau (Fig 4B).

Next, we explore the long-time growth of actin filament within a fixed time window. The mean and the variance of the long-time distribution of $\Delta L_\tau$ are given by

$$\langle \Delta L_\tau \rangle = r_1 A_C \tau, \tag{10}$$

$$\sigma^2_{\Delta L_\tau} = r_1 A_C \tau + \frac{2}{D_C} r_1^2 A_C(1 - A_C)\tau - \frac{2}{D_C^2} r_1^2 A_C(1 - A_C)(1 - e^{-\tau D_C}). \tag{11}$$

The mean long-time growth is highest without capper and decreases with increasing capper concentration (Fig 4C), as capper reduces the time filaments spend in the bare (B) state, where polymerization occurs. The Fano factor remains at one at low capper concentrations but shows non-monotonic behavior: it rises, peaks when B and BC states are equally occupied, and then returns to one (Fig 4C). Over longer time windows, variability increases due to higher overall growth. The convergence of Fano factor to one at both extremes of capper concentration reflects that the dynamics of actin filament is governed by a single state in these limits. When the capper concentration is very low, the filament predominantly remains in the bare (B) state; at very high capper concentrations, it spends most of its time in the capped (BC) state. This limiting behavior is evident from Eqs (10) and (11). When the capper concentration is very low (i.e., $[C] \rightarrow 0$), $D_C \approx k_C^-$ and $A_C \approx 1$. In this limit, the mean filament

growth becomes $\langle \Delta L_\tau \rangle \approx r_1 \tau$, and the variance also becomes $\sigma^2_{\Delta L_\tau} \approx r_1 \tau$, leading to Fano factor $\approx 1$. Similarly, at very high capper concentration, $D_C \gg 1$ and $A_C \ll 1$, due to which the mean growth becomes $\langle \Delta L_\tau \rangle \approx r_1 A_C \tau$ and variance becomes $\sigma^2_{\Delta L_\tau} \approx r_1 A_C \tau$, again giving Fano factor $\approx 1$.

## Combined effects of an elongator and a capper on actin filament length

So far, we have quantified the effects of individual regulatory proteins, such as elongators and cappers, on actin filament length. However, within cells, these factors operate concurrently, often competing for the same binding site on a filament [29,30]. To explore the combined effect of an elongator and a capper, we consider two models, see Fig 5A and 5B. First, Competitive Binding Model: The elongator and capper bind the same filament end in a mutually exclusive manner. In this model, the filament can exist in three states—free (B), capper-bound (BC), or elongator-bound (BF)—with polymerization rates $r_1$, $r_2$, and $r_3 = 0$, respectively (Fig 5A). Second, Simultaneous Binding Model: Both proteins can simultaneously bind the same filament end. Thus, the filament can now occupy four states—free (B, polymerization

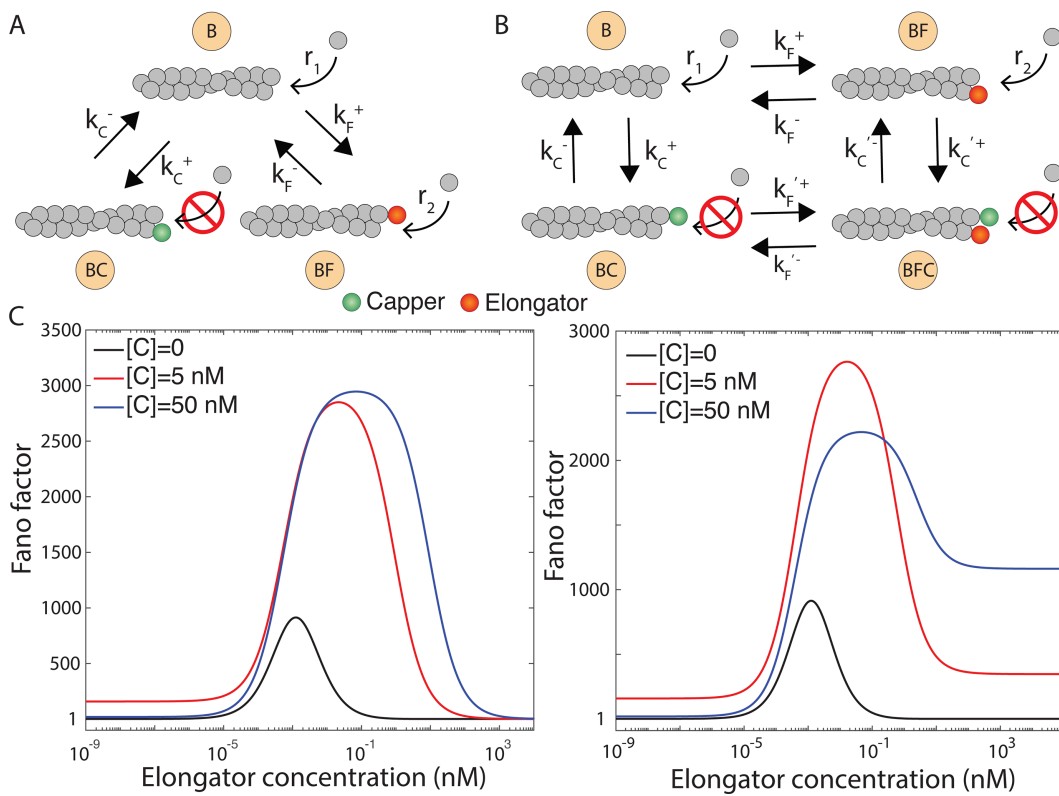

**Fig 5. Combined effect of a capper and an elongator on actin filament length.** (A) Competitive and (B) simultaneous binding models are shown. (C) Fano factor in long-time is shown as a function of elongator concentration for various capper concentrations for competitive (left) and simultaneous (right) binding models. These long-time plots are generated for a fixed time window, $\tau = 100s$. Here, we express Fano factor in the unit of subunits of actin monomers. Following parameters for elongator (formin) and capper (capping protein) are used: $r_1 = 6$ subunits/$s$, $r_2 = 30$ subunits/$s$, $\widetilde{k}_F^+ = 29.1 \ \mu M^{-1}s^{-1}$, $k_F^- = 8.1 \times 10^{-5} \ s^{-1}$, $\widetilde{k}_C^+ = 12.8 \ \mu M^{-1}s^{-1}$ $k_C^- = 2.0 \times 10^{-4} \ s^{-1}$, $\widetilde{k}_F^{'+} = 1.6 \ \mu M^{-1}s^{-1}$, $\widetilde{k}_C^{'+} = 0.21 \ \mu M^{-1}s^{-1}$, and $k_F^{'-} = k_C^{'-} = 6.2 \times 10^{-3} \ s^{-1}$ [23].

rate $r_1$), elongator-bound (BF, polymerization rate $r_2$), capper-bound (BC, polymerization rate $r_3 = 0$), or dual-bound (BFC, polymerization rate $r_4 = 0$) (Fig 5B).

To differentiate between these two models, we analyze the long-time variability in filament growth within a fixed time window by varying elongator concentration at different capper concentrations (Fig 5C). In both models, the Fano factor shows non-monotonic behavior: it rises from low values at low elongator concentrations, followed by a peak at intermediate concentrations, and then decreases at higher concentrations. In the competitive binding model, the Fano factor approaches one at high elongator concentrations, indicating Poissonian growth. In contrast, the simultaneous binding model yields a Fano factor greater than one, which increases with capper concentration. These differences reflect how the state occupancy of the filament changes with elongator concentration. In the competitive model, high elongator concentrations increase the occupancy of the elongator-bound state, reducing variability in filament growth. In the simultaneous binding model, the filament alternates between two states, leading to higher variability even at high elongator concentrations. Such concentration dependence is consistent with previous studies [17,18], which demonstrated that the probability distribution of filament length varies with concentrations of formin and capping protein.

## Discussion

In this work, we have developed a general and analytically tractable kinetic framework for modeling actin filament dynamics under multicomponent regulation. By allowing each filament to stochastically transition between discrete states– each representing different combinations of bound actin-binding proteins (ABPs)– our model captures the coupled effects of multiple regulatory factors on filament polymerization and depolymerization. Unlike previous approaches that typically focus on steady-state behavior or involve intensive stochastic simulations, our framework provides exact analytical expressions for both the time evolution of filament length distributions and the long-time distribution of changes in filament length within a fixed time window. An important implication of our work is that it enables systematic discrimination between different mechanisms of length control of actin filaments based on measurable filament statistics. For example, by employing our framework we find that changes in the concentration of a single ABP can alter the long-time and temporal variance of filament lengths, depending on its kinetic properties and interactions with other regulators. These predictions help interpret existing experimental data and propose new experimental strategies to differentiate between competing regulatory mechanisms.

So far, we have focused on a single elongator and capper. However, cells contain a diverse array of regulatory factors, including depolymerases like twinfilin and cofilin [25], as well as multiple elongators and cappers such as formins [27] (with 15 mammalian isoforms) and Ena/VASP proteins [31]. Additional complexity arises from the presence of two distinct filament ends and filament age, both of which modulate protein binding and activity. To illustrate the broad applicability of our framework, we extended our analysis to a system comprising three regulators—an elongator, a capper, and a depolymerase—as detailed in S1 Text. We also examined analytically tractable benchmark cases, including a one-state depolymerizing system and two-state polymerizing-depolymerizing model. These simplified systems recover well-established behaviors: the one-state model exhibits Poissonian fluctuations (Fano = 1) at both transient and long-time limits, while the two-state model behaves as a symmetric random walk under fast switching and balanced polymerization-depolymerization conditions. These results confirm the applicability of our framework and highlight its flexibility in capturing increasingly complex regulatory architectures, offering a unified approach for dissecting actin filament dynamics in physiological contexts.

In some of our case studies, the model predicts extremely high Fano factors– up to $\sim$ 1000– which reflect large variability in filament lengths. Due to such variability, a statistically systematic comparison between theory and experiments requires a discussion of the experimental sample size. We have derived an expression for the required sample size needed to achieve a target relative error $\epsilon$ in Fano factor estimation (see S1 Text), given by

$$N \sim \frac{1}{\epsilon^2}\left(2 + 0.0027 \times \frac{F}{\mu}\right),$$

where, the Fano factor $F$ is expressed in subunits and the mean filament growth $\mu$ in $\mu m$. This analysis shows that even for large Fano factor values, the required sample size remains experimentally feasible. For instance, even when $F \sim 1000$ and $\mu \sim 10 \mu m$, the expression reduces to $N \sim \left(1/\epsilon^2\right) \times 2$. Thus, for a 10% relative error ($\epsilon = 0.1$), a sample size of $N \sim 200$ would be sufficient. This confirms that the Fano factor values reported in our main text can be estimated with reasonable confidence, reinforcing the practical applicability of our analytical framework.

To further demonstrate the relevance of our theory, we compared theoretical predictions from the two-state elongator model (see Fig 4A) with *in vitro* data on actin filament elongation in the presence of formin, as reported in Shekhar et al. [23]. In this experiment, actin filaments were polymerized in the presence of formin mDia1, and filament lengths were recorded over time using mF-TIRF microscopy [32]. From the published microscopy videos [23], we extracted time-resolved length trajectories for 37 individual actin filaments and calculated the Fano factor of the filament length distribution at each time point. As shown in Fig 6, the

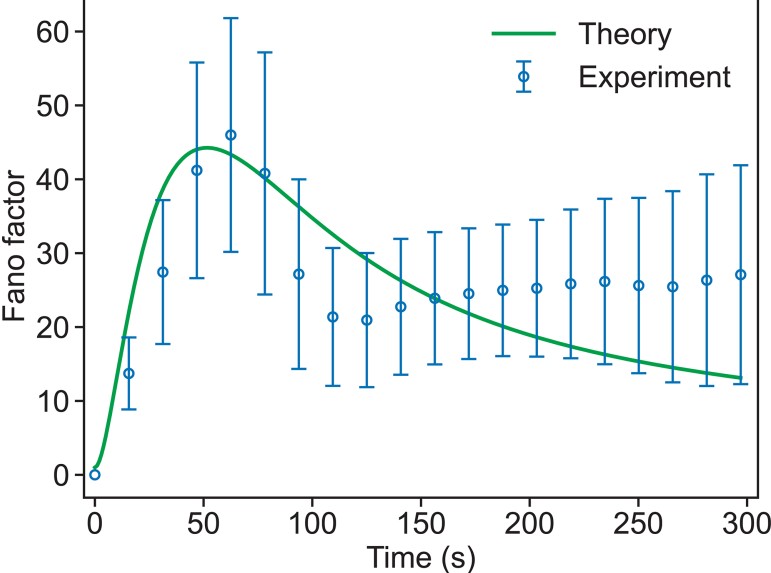

**Fig 6. Comparison of theoretical and experimental Fano factor for actin filament length regulated by formin.**
The theoretical curve is obtained from the two-state model given in Fig 3A. Experimental data are taken from Shekhar et al. [23], where multiple filaments were grown in the presence of of formin in the growth medium. The following parameters are used to generate the theoretical curve that best fits the experimental trend: $r_1$ = 6 subunits/s, $r_2$ = 16 subunits/s, $\widehat{k}_F^+$ = 8.8 $\mu M^{-1}s^{-1}$, $k_F^-$ = 14.7 $\times 10^{-5}$ $s^{-1}$, and $[F]$ = 5 nM. These values are comparable to those reported in [23] and qualitatively reproduce the experimental Fano factor dynamics. Details of the experimental data extraction from TIRF image sequences are provided in S1 Text.

two-state elongator model qualitatively captures the experimental trend, including the non-monotonic behavior of the Fano factor. The agreement is based on a modest dataset of 37 filaments and is accompanied by bootstrap-estimated error bars that reflect the statistical variation across the filament population. For instance, at $t \sim 50s$, the experimental Fano factor is $\sim 45$, with a corresponding error of about $\sim 15$, yielding a relative error of $\sim 33\%$. This uncertainty in our estimate is due to small sample size. The analysis indicates that the two-state elongator model, as a simple case of our general framework, is capable of reproducing filament growth variability observed in experiments. However, more targeted experiments that systematically investigate filament length over time in the presence of different regulatory proteins will be quintessential for testing the broader, fluctuation-based predictions of our general framework.

In summary, our results provide a theoretical framework for interpreting fluctuations in actin filament length under multicomponent regulation. While most existing models focus on specific mechanisms involving a limited number of regulatory proteins [14,15,17–22], our generalized analytical framework captures the collective influence of an arbitrary number of ABPs on filament dynamics. A dialogue between our theoretical framework and experiments that track actin filament lengths over time [11–15] can pave the way for a deeper mechanistic understanding of how multiple regulatory proteins together control filament length in cells.

## Supporting information

**S1 Text. Supporting information.** Includes detailed derivations of statistical moments for the general kinetic model, illustrative examples, procedures for error estimation, and the methodology used for analyzing experimental data. Includes Fig A, showing the schematics of the matrices $\hat{\mathbf{R}}$, $\hat{\mathbf{K}}$, $\mathbf{P}(\Delta L_t)$, and the decomposition of the transition probability vector. (PDF)

## Author contributions

**Conceptualization:** Mintu Nandi, Shashank Shekhar, Sandeep Choubey.

**Formal analysis:** Mintu Nandi, Shashank Shekhar, Sandeep Choubey.

**Funding acquisition:** Mintu Nandi, Shashank Shekhar, Sandeep Choubey.

**Investigation:** Mintu Nandi, Shashank Shekhar, Sandeep Choubey.

**Methodology:** Mintu Nandi, Shashank Shekhar, Sandeep Choubey.

**Project administration:** Shashank Shekhar, Sandeep Choubey.

**Resources:** Mintu Nandi, Shashank Shekhar, Sandeep Choubey.

**Software:** Mintu Nandi.

**Supervision:** Shashank Shekhar, Sandeep Choubey.

**Visualization:** Mintu Nandi, Shashank Shekhar, Sandeep Choubey.

**Writing – original draft:** Mintu Nandi, Shashank Shekhar, Sandeep Choubey.

**Writing – review & editing:** Mintu Nandi, Shashank Shekhar, Sandeep Choubey.

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
