## [Decision Letter · Decision Letter 0]

1 Apr 2025

PCOMPBIOL-D-25-00161

A generalized theoretical framework to investigate multicomponent actin dynamics

PLOS Computational Biology

Dear Dr. Shekhar,

Thank you for submitting your manuscript to PLOS Computational Biology. After careful consideration, we feel that it has merit but does not fully meet PLOS Computational Biology's publication criteria as it currently stands. Therefore, we invite you to submit a revised version of the manuscript that addresses the points raised during the review process.

Please submit your revised manuscript within 60 days Jun 01 2025 11:59PM. If you will need more time than this to complete your revisions, please reply to this message or contact the journal office at ploscompbiol@plos.org. Please include the following items when submitting your revised manuscript:

We look forward to receiving your revised manuscript.

Kind regards,

Christopher E Miles

Academic Editor

PLOS Computational Biology

Mark Alber

Section Editor

PLOS Computational Biology

**Additional Editor Comments:**

Overall, the reviewers found the manuscript to be fairly clear and the work to have strong possibility of impact in quantitative understanding of actin dynamics. The analytical analysis was a notable strength. However, there are major concerns that each reviewer expressed, with major themes including (1) challenges in connecting the theory with data and (2) questions about the modeling assumptions and their precise description. The authors should address these concerns raised by the reviewers as directly as possible for consideration of publication.

**Journal Requirements:**

At this stage, the following Authors/Authors require contributions: Shashank Shekhar. Please ensure that the full contributions of each author are acknowledged in the "Add/Edit/Remove Authors" section of our submission form.

5) We have noticed that you have uploaded Supporting Information files, but you have not included a list of legends. Please add a full list of legends for your Supporting Information files after the references list.

6) Please ensure that the funders and grant numbers match between the Financial Disclosure field and the Funding Information tab in your submission form. Note that the funders must be provided in the same order in both places as well.

**Reviewers' comments:**

Reviewer's Responses to Questions

**Comments to the Authors:**

Reviewer #1: How the multitude of actin binding proteins (ABPs) found in cells contribute to controlling the sizes of actin filaments is an interesting and important question. To address this, the authors present kinetic model of actin filament assembly that considers how these regulatory factors influence both the evolution of filament length over time, as well as the steady-state distribution of filament lengths. They use this model to explore how these aspects of filament length change in the presence of an elongator, a capper, and finally a mixed population of these two regulators. They find that in addition to their influence over the length of filaments, these regulators have different effects on the filament-to-filament variability in length. While this model may present a useful way to distinguish between different regulatory mechanisms of multiple ABPs, there are a number of issues that should be addressed prior to publication.

Major issues:

1. The description of the model is tough to parse and could use a plain-language description of how this model differs from prior models (especially: https://doi.org/10.1016/j.bpj.2022.05.014 and 10.1146/annurev-biophys-070915-094206). As written, it is hard to assess the novelty of this model.

2. While the authors claim that this model can capture the simultaneous effects of an arbitrary number of ABPs they only explore scenarios that include either single regulators or a pair of regulators. There are many examples of models that consider the effect of 1-2 regulators, so it would be nice to have a demonstration of how this model can be applied to 2+ regulators.

3. The examples used do not appear to include filament disassembly and only consider filaments that have the capacity to grow. Can the authors comment why disassembly is not included in these examples?

4. In the description of the model the authors mention that capped filament states can be considered as part of the state leading to polymerization. These seems counterintuitive and this assumption could use further discussion.

5. In figure 3B, the x-axis in the right panel is truncated compared to the panel on the left. Extending this axis would support the author’s claim that the fano factor reaches a plateau in the long-time limit. Currently, the lower concentration of CP would appear to increase.

6. The results from Figure 4 appear to be the strongest support for their claims. Do the results in 4C support prior models of formin/capping protein behavior where the distribution depends on the concentration of these regulators?

Minor issues:

1. The manuscript should be carefully reviewed as there are a number of typos. Without line numbers it is difficult to point out specific examples.

2. The arrows in the cartoon of Fig 2 appear to be mislabeled.

3. A clear definition of what ‘mean filament growth’ means and whether this different than ‘length’ would be helpful.

4. Figure 4A – BF and BC are mislabeled, as are the arrows.

5. Figure 4C is not referenced in the manuscript and the figure legend should mention which plot corresponds to which model.

Reviewer #2: see attached

Reviewer #3: Please find our feedback in the uploaded attachment.

**Have the authors made all data and (if applicable) computational code underlying the findings in their manuscript fully available?**

Reviewer #1: None

Reviewer #2: None

Reviewer #3: **No: **Not applicable.

PLOS authors have the option to publish the peer review history of their article (what does this mean?). If published, this will include your full peer review and any attached files.

Reviewer #1: No

Reviewer #2: No

Reviewer #3: No

**Figure resubmission:**
---

## [Decision Letter · Decision Letter 1]

13 Aug 2025

Dear Dr Shekhar,

We are pleased to inform you that your manuscript 'A generalized theoretical framework to investigate multicomponent actin dynamics' has been provisionally accepted for publication in PLOS Computational Biology.

Best regards,

Christopher E Miles

Academic Editor

PLOS Computational Biology

Mark Alber

Section Editor

PLOS Computational Biology

Reviewer's Responses to Questions

**Comments to the Authors:**

Reviewer #1: The revised manuscript adequately addresses my prior concerns and has substantially improved.

Reviewer #2: The authors have adequately addressed my comments from the previous round of review.

Reviewer #3: We thank the authors for their careful statistical and experimental work to address our comments. We have no further questions.

**Have the authors made all data and (if applicable) computational code underlying the findings in their manuscript fully available?**

Reviewer #1: None

Reviewer #2: None

Reviewer #3: Yes

PLOS authors have the option to publish the peer review history of their article (what does this mean?). If published, this will include your full peer review and any attached files.

Reviewer #1: No

Reviewer #2: No

Reviewer #3: No

---

## [Editor Report · Acceptance letter]

PCOMPBIOL-D-25-00161R1

A generalized theoretical framework to investigate multicomponent actin dynamics

Dear Dr Shekhar,

I am pleased to inform you that your manuscript has been formally accepted for publication in PLOS Computational Biology. Your manuscript is now with our production department and you will be notified of the publication date in due course.

With kind regards,

Benedek Toth
